# The incidence of and risk factors for late presentation of childhood chronic kidney disease: A systematic review and meta-analysis

Lucy Plumb[1,2]*, Emily J. Boother[3], Fergus J. Caskey[1,4], Manish D. Sinha[5,6], Yoav Ben-Shlomo[1,7]

**1** Population Health Sciences, University of Bristol Medical School, Bristol, United Kingdom, **2** UK Renal Registry, The Renal Association, Bristol, United Kingdom, **3** Faculty of Medicine, Imperial College London, London, United Kingdom, **4** Department of Renal Medicine, North Bristol NHS Trust, Bristol, United Kingdom, **5** Department of Paediatric Nephrology, Evelina London Children's Hospital, Guys and St Thomas' NHS Foundation Trust, London, United Kingdom, **6** King's British Heart Foundation Centre, King's College London, London, United Kingdom, **7** The National Institute for Health Research Applied Research Collaboration West (NIHR ARC West), University Hospitals Bristol and Weston NHS Foundation Trust, Bristol, United Kingdom

* lucy.plumb@bristol.ac.uk

**Data Availability Statement:** All relevant data are within the manuscript and its supporting information files.

## Abstract

### Background

When detected early, inexpensive measures can slow chronic kidney disease progression to kidney failure which, for children, confers significant morbidity and impacts growth and development. Our objective was to determine the incidence of late presentation of childhood chronic kidney disease and its associated risk factors.

### Methods

We searched MEDLINE, Embase, PubMed, Web of Science, Cochrane Library and CINAHL, grey literature and registry websites for observational data describing children <21 years presenting to nephrology services, with reference to late presentation (or synonyms thereof). Independent second review of eligibility, data extraction, and risk of bias was undertaken. Meta-analysis was used to generate pooled proportions for late presentation by definition and investigate risk factors. Meta-regression was undertaken to explore heterogeneity.

### Results

Forty-five sources containing data from 30 countries were included, comprising 19,339 children. Most studies (37, $n = 15,772$) described children first presenting in kidney failure as a proportion of the chronic kidney disease population (mean proportion 0.43, 95% CI 0.34–0.54). Using this definition, the median incidence was 2.1 (IQR 0.9–3.9) per million age-related population. Risk associations included non-congenital disease and older age. Studies of hospitalised patients, or from low- or middle-income countries, that had older study populations than high-income countries, had higher proportions of late presentation.

**Funding:** L.P is funded by the National Institute for Health Research (NIHR) Research Trainees Coordinating Centre, (Doctoral Research Fellowship 2016-09-055) for this research project. This publication presents independent research funded by the National Institute for Health Research (NIHR). The views expressed are those of the authors and not necessarily those of the NHS, the NIHR or the Department of Health and Social Care. The funder had no role in study design, data collection and analysis, decision to publish, or preparation of the manuscript.

**Competing interests:** L.P reports grants from Kidney Research UK during the conduct of the study. F.J.C reports grants from NIHR, grants from Kidney Research UK, and personal fees from Baxter outside the submitted work. M.D.S acknowledges financial support from the Department of Health via the National Institute for Health Research (NIHR) comprehensive Biomedical Research Centre and Clinical Research Facilities awards to Guy's and St Thomas' NHS Foundation Trust in partnership with King's College London and King's College Hospital NHS Foundation Trust. Y.B-S is partly funded by National Institute for Health Research Applied Research Collaboration West (NIHR ARC West) at University Hospitals Bristol and Weston NHS Foundation Trust. This does not alter our adherence to PLOS ONE policies on sharing data and materials.

## Conclusions

Late presentation is a global problem among children with chronic kidney disease, with higher proportions seen in studies of hospitalised children or from low/middle-income countries. Children presenting late are older and more likely to have non-congenital kidney disease than timely presenting children. A consensus definition is important to further our understanding and local populations should identify modifiable barriers beyond age and disease to improve access to care.

## Introduction

Chronic kidney disease (CKD) in children and young people is a significant health problem associated with impaired growth, bone disease, cardiovascular morbidity, and a reduced life expectancy [1–3]. Furthermore, poorer health-related quality of life and psychosocial functioning is seen compared to healthy controls, which extends into early adulthood [4–6]. Although data are limited, population-based studies suggest up to 1% of the child population may have reduced kidney function [7]. Progression to kidney failure requires life-long kidney replacement therapy (KRT) as dialysis or kidney transplantation for survival, which is costly and unaffordable for many health-care budgets worldwide. A recent global systematic review estimated that while 2.6 million people received KRT in 2010, this was likely represented only half of all those requiring life-sustaining treatment [8]. Paediatric-specific estimates suggest that the median incidence of KRT-treated ESKD is 9 (range 4–18) cases per million of the age-related population [9].

International efforts are progressively directed towards timely identification of CKD given the increasing appreciation of its contribution to the global burden of non-communicable disease. When detected early, inexpensive measures can delay or slow disease progression to kidney failure, thus mitigating costs associated with KRT provision [10]. Implementing these strategies however is dependent both on timely recognition of CKD and appropriate access to healthcare. For many children, diagnosis of CKD occurs when kidney function is already severely reduced [11], although the reasons for this are not fully understood. In adults, risk factors for late presentation include older age, multi-morbidity, Black or Hispanic ethnicity, and being uninsured; lack of communication between primary and specialist care can also influence timing of referral [12]. These results may not be generalizable to children given differences in kidney disease aetiology and management.

Our aim was to conduct a systematic review of the epidemiological and clinical literature to determine the incidence and proportion of late presentation of childhood CKD and associated risk factors.

## Materials and methods

We performed a systematic review to identify studies describing the proportion (within the clinical population) or incidence (with reference to the general population) of children presenting late to specialist or hospital services with CKD. A review protocol was registered with an international systematic review database (www.crd.york.ac.uk/prospero, reference: CRD42017064098). Findings are reported in accordance with the Preferred Reporting Items for Systematic Reviews and Meta-Analyses checklist [13].

Studies that described first presentation of incident children with CKD and/or kidney failure, with reference to late or delayed presentation/referral (or synonyms thereof) were

included. As there is no consensus definition for late presentation in children, all published definitions were accepted. Similarly, studies describing paediatric populations aged up to 21 years were included. Comparable studies describing stage of kidney function at presentation or diagnosis that did not reference late presentation were also included. Reports focusing on specific disease types or age-groups (e.g. adolescents) were excluded.

With support from an experienced information specialist, a comprehensive search strategy was developed (S1 Fig) and adapted to search six databases: Ovid MEDLINE, EMBASE, CINAHL, PubMed, the Cochrane library, and Web of Science (science and social science citation indices). The first search was performed in September 2017, which was repeated in 2019 (results limited to the previous 2-years). Forward- and backward-reference searches were undertaken. Five grey literature databases were searched (S2 Fig). Kidney registry websites with good public accessibility were also examined for relevant reports [14]. We included all non-English studies, although only English grey literature was explored. Data available in published abstracts form only was excluded. Studies published between 1 January 1990 and 10 September 2019 which were available in paper or electronic form were included: this time-frame was chosen to adequately capture studies from established paediatric nephrology centres and reflect modern specialist healthcare access globally.

References were imported into EndNote™ which was used to remove duplicates. L.P screened titles and abstracts of all studies to identify those meeting the pre-determined eligibility criteria. E.J.B screened a proportion (20%) of the original search, with 99.9% agreement between reviewers. Any differences were resolved through discussion. Google Translate was used to determine eligibility of non-English studies; where necessary, formal full-text translation was undertaken. We sought to contact authors where clarity was required over the incident or prevalent nature of the study cohort. For registry data, the most comprehensive data available, whether from annual report or publication, were used.

A data extraction proforma was devised and piloted prior to use. For studies reporting all incident kidney disease, numbers of children with CKD were used as the denominator population to ensure comparability across studies. A second reviewer (E.J.B) performed independent data extraction for 20% of studies from the original search. There was 100% agreement between data extractors.

A modified Newcastle-Ottawa scale (NOS) [15] was devised to quantify risk of bias relevant to the study aim, which may have contrasted from the study's original purpose (S1 Table). L.P performed a risk of bias assessment for all included studies; independent second review (E.J.B) was undertaken for 20% of studies from the original search, with disagreements resolved through discussion.

## Data analysis

A descriptive analysis was undertaken for included studies according to late presentation definition used. The terminology contained within this report aligns with recommendations from Kidney Disease Improving Global Outcomes (KDIGO) consensus conference on nomenclature for kidney function and disease [16], although reference to study-specific terminology is made. For studies containing population data, the annual incidence of late presentation was calculated per million of the age-related population (pmarp). Where possible, meta-analysis of the proportion of LP for individual studies was performed by study definition, using a random effects (DerSimonian-Laird) model due to expected high heterogeneity among studies. Score confidence intervals (CI) and Freeman-Tukey double arcsine transformation were used to ensure admissible confidence intervals for individual and pooled proportions respectively [17]. Pooled risk ratios (RR) from meta-analysis of clinical and demographic data were

calculated to explore risk associations with late presentation. Publication bias was evaluated using a funnel plot, Begg's test and Egger's linear regression. All analyses were conducted using Stata v15.

## Results

We identified 49 studies meeting inclusion criteria for our systematic review. Fig 1 outlines the result of our searches, with reasons for exclusion at each stage. Of the eligible studies, 4 were removed from further analysis [18–20], leaving 45 studies (S2 Table). Five late presentation descriptions were identified (Table 1).

### Group one (37 studies): Advanced kidney disease/kidney failure at first presentation (denominator: CKD population [21–57]

Group 1 studies comprised 15,772 participants with a median sample size of 139 (interquartile range, IQR 48–305, Table 2). Most studies were small, single-centre and from low- or middle-income countries (LMIC, 75.7%). Functional thresholds for inclusion differed across studies: most common was an estimated glomerular filtration rate (eGFR) of $<60ml/min/1.73m^2$ used in 10 studies (27.0%). The upper eGFR inclusion threshold for remaining studies ranged from 25 to $>90$ ml/min/1.73m$^2$. Four studies used serum creatinine levels to determine eligibility; 1 study did not report a threshold value.

Within this group, variations in the description of late presentation were also noted. The term 'kidney failure' or 'end-stage kidney disease' itself carried different meanings: 18 studies referred to kidney failure as an eGFR of $<15ml/min/1.73m^2$ (48.6%); in 9 studies (24.3%) a threshold of $<10$ ml/min/1.73m$^2$ was used. Two studies did not offer a value or reference, while 1 study defined kidney failure as requiring KRT or death from kidney failure. Five studies included stage 4 CKD in reference to late presentation while 2 described children with an eGFR at presentation of $<25$ ml/min/1.73m$^2$.

### Group two (8 studies): First presentation in kidney failure and/or requirement for KRT within specified time-frame (denominator: Incident kidney failure population) [11,58–64]

Group 2 studies included 3,567 participants with a median sample size of 146 (IQR 50–779, Table 3). Compared with group 1, group 2 studies were more likely to come from high-income countries (HIC, $p = 0.01$); half were multi-centre in design with data from 5 studies (62.5%) obtained from a registry dataset. Seven studies included in this group used a requirement for KRT as a means of identifying the study cohort, with 1 study including children who died from kidney failure. One study described children presenting to nephrology care, all of whom developed kidney failure within a 6-month time-frame and was therefore included in this group. Again, within this group differences in time-based definitions were noted: 3 studies used a 3-month time-frame between presentation and KRT start to define late presentation [11,58,59]; 2 studies used 1 month [60,61]; 1 study described a 6-month time-frame [62]. Two studies described the proportion of children not known to nephrologists before starting KRT [63,64].

### Group three (3 studies): eGFR-based threshold definition (incident kidney failure population) [11,58,61]

Three studies [11,58,61] from Australia, UK and Europe respectively, offered additional data describing the proportion of incident children with kidney failure with a low eGFR at

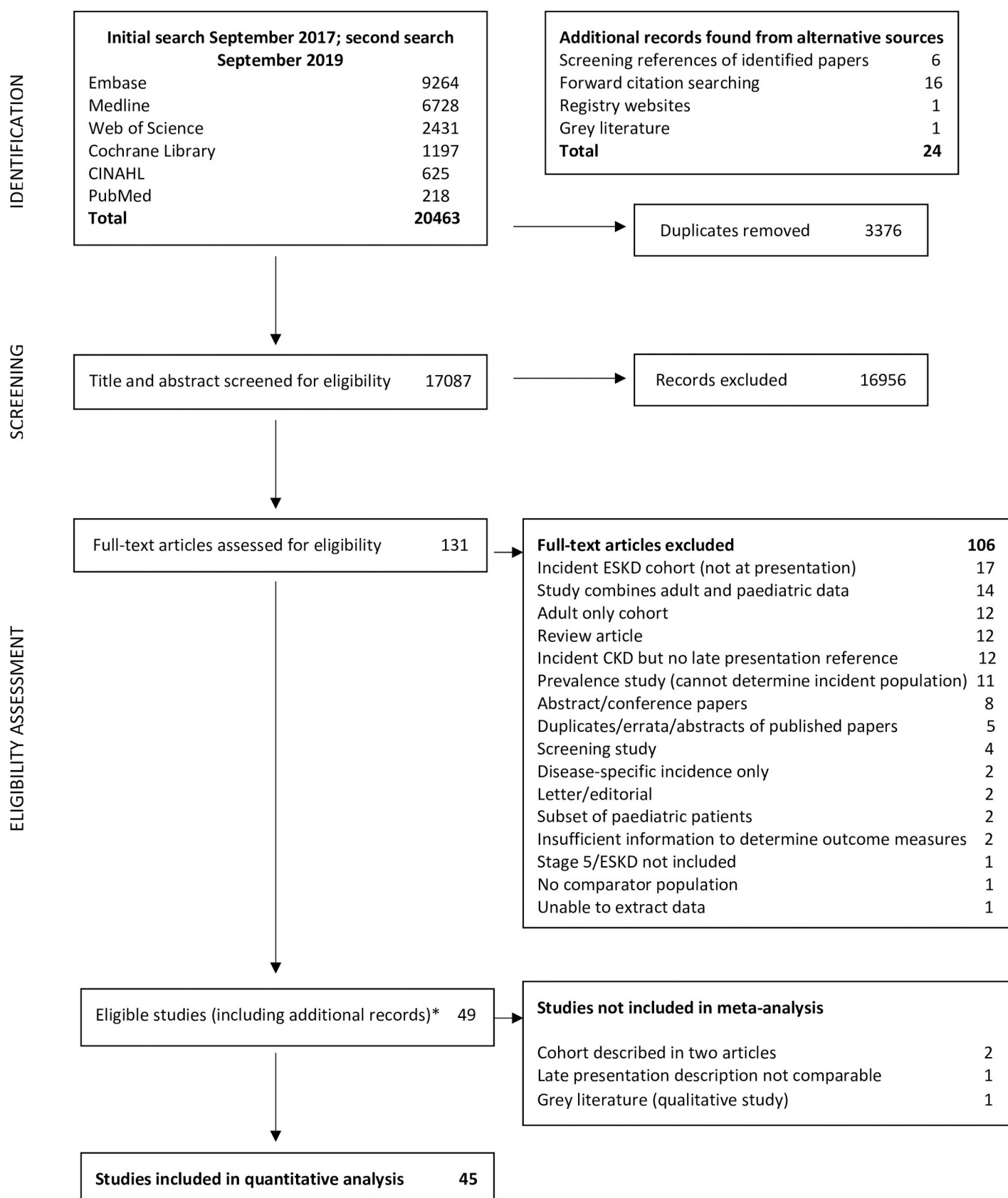

**Fig 1. PRISMA diagram.** *24 papers were identified from additional sources. Abbreviations: CKD, Chronic Kidney Disease; ESKD, End-stage Kidney Disease (term used synonymously with 'kidney failure').

presentation to nephrology services. All were retrospective analyses of registry data, 2 were multi-centre in design. Two studies [11,58] used <30ml/min/1.73m$^2$ while 1 study [61] used ≤20 ml/min/1.73m$^2$ to define low kidney function at first review.

## Group four (2 studies): Time delay between symptom onset and diagnosis of CKD >3 months [24,44]

Two Nigerian, single-centre studies (252 children, 55.1% male) described the proportion of incident children who experienced symptoms for >3 months before attendance. The study cohort was identified from paediatric nephrology referrals in 1 and hospital admissions in another. Both were retrospective in design and included children aged ≤16 years. Another Nigerian single-centre study reported a mean duration of symptoms of 21.2 (standard deviation 5.8) weeks [31]; due to a lack of comparable data, this study was not included.

## Group five (2 studies): Time delay between presentation and paediatric nephrology referral [21,22]

Two single-centre, retrospective studies with a total sample size of 176 patients (67.0% male), described time delays to nephrology referral. This was evidenced by disparities in mean age at initial presentation and subsequent paediatric nephrology referral, with differences of 1.7 and 2.1 years reported.

The median modified NOS percentage score for published studies was 75.0% (IQR 75.0–100.0). Study populations were commonly incident patients attending single-centre paediatric nephrology services, although 9/45 studies (20.0%) sampled hospital admissions only and therefore may not be representative of all children presenting to specialist care. Most registry-based studies relied on voluntary reporting of cases which raises the possibility of under-representation. Furthermore, reporting of exclusion criteria was limited and varied widely (S2 Table), restricting assessment of selection bias. Four studies were identified as having an analytic component relevant to our study aim however only 2 adjusted for potential confounders, leaving the remainder open to confounding bias. All but 2 studies specified use of data collated

**Table 1. Late presentation definitions identified in systematic review.**

| Study group | Definition | Number of studies | Total sample size |
|---|---|---|---|
| 1 | Advanced kidney disease/ESKD/kidney failure at first presentation (denominator: CKD population) | 37 | 15772 |
| 2 | First presentation in ESKD/kidney failure and/or requirement for KRT within specified timeframe (denominator: incident ESKD/kidney failure population) | 8 | 3567 |
| 3 | Estimated glomerular filtration rate-based threshold (denominator: incident ESKD/kidney failure population) | 3 | 1427 |
| 4 | Time delay between symptom onset and diagnosis of CKD >3 months | 2 | 252 |
| 5 | Time delay between secondary and tertiary care referral | 2 | 176 |

Abbreviations: CKD, Chronic Kidney Disease; ESKD, End-Stage Kidney Disease (used synonymously with 'kidney failure').

**Table 2. Group one: Study characteristics and findings (Group 1, *n* = 37) [21–57].**

| Study characteristics | Number (%) | Study findings | Number (%) |
|---|---|---|---|
| **Study location:** | | **Number of participants** | |
| Asia | 19 (51.4) | Range | 3–6969 |
| Africa | 7 (18.9) | Mean (SD) | 426 (1153) |
| Europe | 6 (16.2) | Median (IQR) | 139 (48–305) |
| North America | 5 (13.5) | | |
| Low-middle income country | 28 (75.7) | | |
| **Year(s) of study:** | | **Sex: n (%)** *n = 29 studies*: | |
| Range | 1990–2014 | Male | 4092 (60.9) |
| Median year of publication (IQR) | 2010 (2005–2014) | Female | 2628 (39.1) |
| **Duration of study (years):** | | **Age of participants:** | |
| Median (IQR) | 6.0 (3.0–10.0) | Mean age (SD) *n = 18 studies* | 8·1 (2.7) |
| Range | 1.0–17.0 | Median age (IQR) *n = 11 studies* | 9 (5.6–12.5) |
| **Sampling method:** | | **Kidney function at first presentation:** | |
| Single centre | 26 (70.3) | Mean eGFR (SD) *n = 10 studies* | 28.2 (18.5) |
| Multi-centre | 11 (29.7) | Median eGFR (IQR) *n = 5 studies* | 18.5 (14.9–30.5) |
| **Studies using registry data** | 6 (16.1) | | |
| **Age inclusion** | | **Late presentation definitions used** | |
| <12 years | 2 (5.4) | Stage 4 or 5 CKD | 5 (13.5) |
| <13 years | 1 (2.7) | eGFR <25 ml/min/1.73m$^2$ | 2 (5.4) |
| <14 years | 1 (2.7) | Stage 5 (eGFR <15) | 18 (48.6) |
| <15 years | 3 (8.1)) | Creatinine clearance/eGFR <10 | 9 (24.3) |
| <16 years | 6 (16.2) | 'End-stage kidney disease'–value not reported | 2 (5.4) |
| <17 years | 6 (16.2) | Requiring KRT or death | 1 (2.7) |
| <18 years | 3 (8.1) | | |
| <19 years | 10 (27.0) | | |
| <20 years | 4 (10.8) | | |
| <21 years | 1 (2.7) | | |
| **CKD inclusion criteria** | | **Modified NOS Score*** | |
| Any CKD stage (1–5) | 3 (8.1) | Median score (max possible score 4 points) *n = 36 studies* | 3 (3.0–4.0) |
| eGFR <90ml/min/1.73m$^2$ | 5 (13.5) | Median percentage score (IQR) | 75 (75.0–100.0) |
| eGFR <75 ml/min/1.73m$^2$ | 4 (10.8) | | |
| eGFR <60 ml/min/1.73m$^2$ | 10 (27.0) | | |
| eGFR <50 ml/min/1.73m$^2$ | 7 (18.9) | | |
| eGFR <25 ml/min/1.73m$^2$ | 1 (2.7) | | |
| Clinical assessment only | 2 (5.4) | | |
| Creatinine based criteria | 4 (10.8) | | |
| Not reported | 1 (2.7) | | |
| **CKD duration threshold** | | | |
| No duration reported | 16 (43.2) | | |
| 3 months/less if clinical evidence | 4 (10.8) | | |
| ≥3 months | 15 (40.5) | | |
| 6 months | 1 (2.7) | | |
| 12 months | 1 (2.7) | | |
| **Guidelines for CKD staging used:** | | | |
| NKF-K/DOQI | 16 (43.2) | | |
| KDIGO | 4 (10.8) | | |
| Textbook definition used | 1 (2.7) | | |

*(Continued)*

**Table 2.** (Continued)

| Study characteristics | Number (%) | Study findings | Number (%) |
|---|---|---|---|
| Not reported | 15 (40.5) | | |
| Country-specific guidelines | 1 (2.7) | | |

Abbreviations: CKD, Chronic Kidney Disease; eGFR, estimated Glomerular Filtration Rate; IQR, Interquartile Range; KDIGO, Kidney Disease Improving Global Outcomes; KRT, Kidney Replacement Therapy; NKF-K/DOQI, National Kidney Federation- Kidney Disease Outcomes Quality Initiative; SD, Standard Deviation.

*Does not include annual report from the North American Pediatric Renal Trials and Collaborative Studies group.

**Table 3. Summary of study population characteristics for studies reporting first presentation in end-stage kidney disease and/or requirement for KRT within specified timeframe (denominator: Incident ESKD/kidney failure population) (Group 2, *n* = 8) [11,58–64].**

| Study characteristics | Number (%) | Study findings | Number (%) |
|---|---|---|---|
| **Study location:** | | **Number of participants** | 3567 |
| Africa | 2 (25.0) | Range | 15–1603 |
| Australasia | 1 (12.5) | Mean (SD) | 446 (615) |
| North America | 1 (12.5) | Median (IQR) | 146 (50–779) |
| Europe | 4 (50.0) | | |
| Low-middle income country[1] | 2 (28.6) | | |
| **Year(s) of study:** | | **Sex:** *n = 5 studies* | |
| Range | 1978–2012 | Male | 1295 (56.9) |
| Median year of publication (IQR) | 2010 (2007–2013) | Female | 979 (43.1) |
| **Duration of study (years):** | | **Age of participants:** | |
| Median (IQR) | 7.0 (3.5–14.6) | Mean age (SD) *n = 3 studies* | 10.9 (3.0) |
| Range | 1.0–28.0 | Median age (IQR) *n = 2 studies* | 7.8 (4.1–11.5) |
| **Sampling method:** | | **Kidney function at first presentation:** | |
| Single centre | 4 (50.0) | Median eGFR *n = 1 study* | 14.8 |
| Multi-centre | 4 (50.0) | | |
| **Studies using registry data** | 5 (62.5) | | |
| **Age inclusion** | | **Late presentation definitions used** | |
| <16 years | 2 (25.0) | Not known to nephrology services before reaching kidney failure/ESKD | 2 (25.0) |
| <18 years | 4 (50.0) | Referral within 1 month of KRT start | |
| <19 years | 1 (12.5) | Referral <3 months (90 days) before KRT start | 2 (25.0) |
| <21 years | 1 (12.5) | Development of kidney failure/ESKD within 6 months of first presentation | 3 (37.5) |
| **Eligibility threshold** | | | |
| Need for KRT or death from kidney failure | 7 (87.5) | | |
| Not recorded | 1 (12.5) | | 1 (12.5) |
| **CKD duration threshold** | | **Modified NOS Score*** | |
| No duration reported | 7 (87.5) | Median score for descriptive studies (max possible score 4 points) *n = 3 studies* (IQR) | 3.0 (2.0–3.5) |
| ≥ 3 months/less if clinical evidence | 1 (12.5) | Median score for analytic studies (max possible score 6 points) *n = 4 studies* (IQR) | 5 (3.8–6.0) |
| | | All studies: median percentage score (IQR) | 75.0 (58.3–100.0) |

Abbreviations: CKD, chronic kidney disease; ESKD, End-Stage Kidney Disease (used synonymously with 'kidney failure'); IQR, interquartile range; KRT, Kidney Replacement Therapy; SD, Standard Deviation.

*Does not include annual report from the US Renal Data System.

[1]Does not include multi-centre European paper in analysis.

from medical or registry records to identify cases, minimising the risk of information bias. Two studies offered no diagnostic criteria which may cause misclassification bias.

Table 4 provides the pooled and individual effect estimates (proportions) of late presentation for studies by definition-group. Group 5 studies which used summary age measures to highlight referral delays were excluded. Overall, a high levels of heterogeneity were noted ($I^2$ >95%). Population figures to calculate incidence were available for 15 group 1 studies covering 14 regions: 2 studies [35,37] described coverage of 1 geographical area (Vietnam) and were combined. The median annual incidence of late presentation was 2.1 (IQR 0.9–3.9) pmarp. A sensitivity analysis excluding studies at high risk of bias (NOS percentage score below median) did not significantly alter pooled proportions overall or by definition-group. Additionally, restricting data to the most recent study from each centre or region did not significantly alter findings.

Sex [11,23,58,59,63] and disease [11,23,25,51,63] data were available from 5 studies, while 4 contained mean age data [23,58,59,60] (Figs 2–4). No association was seen between sex and late presentation (RR 0.89, 95% CI 0.65, 1.21, $p$ = 0.46). Congenital anomalies of the kidneys and urinary tract were associated with a reduced risk of LP (RR 0.60, 95% CI 0.44, 0.82, $p$ = 0.001), although due to heterogeneity in disease groupings, this could not be analysed in further detail. There was strong evidence for a difference in age at presentation (standardised mean difference 0.47 years, 95% CI 0.22, 0.72 $p$<0.001), with a higher mean age noted among late presenting children.

Two studies provided unadjusted data for other risk factors including low socio-economic status (SES) and geographical remoteness respectively. Michael and Gabreil [31] described a cohort of children all presenting with an eGFR of <25ml/min/1.73m$^2$ of which 87.5% were of low SES, although the definition and measure of SES used was unavailable. Kennedy *et al.* reported 36% of late presenting children lived in remote regions compared to 6% of those with timely presentation ($p$ = 0.03) [58]. Qualitative text analysis was undertaken to identify features described but not substantiated that may be linked to late presentation; these included low socioeconomic status and costs associated with healthcare [21,33,36,44] living in remote areas [33], delayed referral to tertiary care [45], 'traditional habits' of first seeking healthcare from traditional healers and a lack of specialist education in local hospitals [37].

Begg's test suggested no evidence ($p$ = 0.53) however Egger's test showed strong evidence of publication bias (coefficient -1.17, $p$<0.001). This was supported by funnel plot asymmetry (S3 Fig) with an over-representation of studies with higher proportions of late presentation.

A meta-regression was performed to determine if the high heterogeneity could be explained by study or ecological covariates. Analysis was restricted to groups 1 and 2. For each study, odds of late presentation were converted to a log-odds scale and a standard error of the log odds calculated. Where studies contained zero-event cells, a continuity correction of 0.5 was added to the number of events and non-events and 1 added to the total study size, to enable inclusion. Covariates of importance were determined *a priori* and included study-level (e.g. group definition, study design (single- or multi-centre), setting (all nephrology referrals versus hospital admissions only)) and ecological variables including summary measures of age and presenting eGFR. Income (LMIC versus HIC) of the study country, as defined by the World Bank [65] was also examined. For continuous variables, median values were preferred; however, to improve power and given similarities in regression coefficients, mean values were included where data were missing.

Table 5 highlights meta-regression findings. Country income, study setting, and age of study cohort were predictive of heterogeneity on univariable analysis. A strong correlation, however, was noted between age and country income, with a lower age at presentation among HIC studies. It was hypothesised that age may mediate the association between country

**Table 4. Late presentation proportion and incidence rates by study definition.**

| Author | Country | Study age threshold | Study kidney function threshold | Late presentation definition used | Late presentation proportion, (95% CI) | % Weight* | Incidence (per million age related population) |
|--------|---------|---------------------|--------------------------------|-----------------------------------|---------------------------------------|-----------|----------------------------------------------|
| **Advanced kidney disease/ESKD at first presentation (denominator: CKD population)** | | | | | | | |
| Mattoo, 1990 | Saudi Arabia | <14 years | Creatinine >180μmol/L (2mg/dL) | Requiring KRT or death | 0.35 (0.26, 0.45) | 2.78 | |
| Al Harbi, 1997 | Saudi Arabia | <13 years | Creatinine >177 μmol/L (2mg/dL) | CrCl <10 | 0.37 (0.26, 0.50) | 2.73 | 9.2 |
| Gulati, 1999 | India | <17 years | Creatinine >180μmol/L (2mg/dL) | ESKD (value not given) | 0.54 (0.40, 0.67) | 2.70 | 0.4 |
| Hafeez, 2002 | India | <16 years | eGFR <75 | eGFR <10 | 0.60 (0.44, 0.73) | 2.68 | |
| Ardissino, 2003 | Italy | <20 years | eGFR <75 | CrCl <25 | 0.26 (0.24, 0.29) | 2.85 | |
| Olowu, 2003 | Nigeria | <16 years | eGFR <60 | eGFR <10 | 0.33 (0.17, 0.55) | 2.52 | 1.1 |
| Hari, 2003 | India | <19 years | eGFR <50 | eGFR <10 | 0.30 (0.25, 0.35) | 2.83 | |
| Michael, 2004 | Nigeria | <17 years | eGFR <25 | eGFR <25 | 1.00 (0.86, 1.00) | 2.56 | |
| Yang, 2004 | China | <15 years | eGFR <50 | Stages 4/5 CKD | 0.81 (0.78, 0.83) | 2.85 | |
| Saeed, 2005 | Syria | <16 years | eGFR <50 | eGFR <10 | 0.61 (0.39, 0.80) | 2.48 | |
| Kari, 2006 | Saudi Arabia | <15 years | eGFR <50 | eGFR <10 | 0.52 (0.40, 0.63) | 2.74 | |
| Hassan, 2007 | Iraq | <18 years | eGFR <50 | eGFR <10 | 0.36 (0.28, 0.46) | 2.78 | |
| Mong Hiep, 2008 | Vietnam | <19 years | Not reported | eGFR <15 | 0.85 (0.81, 0.89) | 2.83 | 2.3^ |
| NAPRTCS, 2008 | United States | <21 years | eGFR <75 | eGFR <15 | 0.12 (0.11, 0.13) | 2.85 | 0.7 |
| Miller, 2009 | Jamaica | <12 years | eGFR <50 | ESKD (value not given) | 0.28 (0.12, 0.51) | 2.48 | 1.4 |
| Bek, 2009 | Turkey | <19 years | eGFR <75 | eGFR <15 | 0.33 (0.27, 0.38) | 2.83 | 3.6 |
| Huong, 2009 | Vietnam | <18 years | Creatinine >150μmol/L (1.7 mg/dL) | eGFR <15 | 0.65 (0.57, 0.72) | 2.80 | |
| Ahmadzadeh, 2009 | Iran | <17 years | eGFR <60 | eGFR <10 | 0.22 (0.16, 0.30) | 2.80 | |
| Mong Hiep, 2010 | Belgium | <20 years | eGFR <60 | eGFR <15 | 0.14 (0.09, 0.21) | 2.80 | 1.7 |
| Peco-Antic, 2011 | Serbia | <19 years | eGFR <90 | eGFR <15 | 0.29 (0.24, 0.34) | 2.83 | 5.8 |
| Gheissari, 2012 | Iran | <19 years | eGFR <60 | eGFR <15 | 0.74 (0.69, 0.79) | 2.82 | |
| Paripović, 2012 | Serbia | <19 years | eGFR <90 | Stages 4/5 | 0.31 (0.23, 0.41) | 2.77 | |
| Olowu, 2013 | Nigeria | <17 years | All CKD stages (1–5) | eGFR <15 | 0.19 (0.13, 0.26) | 2.80 | |
| Kim, 2013 | United Kingdom | <16 years | eGFR <60 | eGFR <15 | 0.14 (0.09, 0.21) | 2.80 | 2.3 |
| Alsaggaf, 2013 | Saudi Arabia | <17 years | eGFR <90 | eGFR <15 | 0.34 (0.29, 0.40) | 2.83 | |
| Keita, 2014 | Senegal | <16 years | eGFR <60 | eGFR <15 | 0.72 (0.58, 0.82) | 2.71 | |
| Odetunde, 2014 | Nigeria | <17 years | eGFR <60 | Stages 4/5 | 0.45 (0.35, 0.55) | 2.77 | 0.8 |
| Cerón, 2014 | Guatemala | <20 years | eGFR <90 | eGFR <15 or KRT | 0.36 (0.32, 0.41) | 2.84 | 2.3 |
| Safouh, 2015 | Egypt | <20 years | All CKD stages (1–5) | eGFR <15 | 0.58 (0.55, 0.61) | 2.85 | 6.5 |
| Kari, 2015 | Saudi Arabia | <15 years | All CKD stages (1–5) | eGFR <15 | 0.16 (0.13, 0.18) | 2.85 | |
| Montini, 2016 | Nicaragua | <19 years | All CKD stages (1–5) | Stages 4/5 | 0.55 (0.49, 0.61) | 2.82 | 4.2 |
| Miller, 2016 | Jamaica | <12 years | eGFR <60 | eGFR <15 | 0.00 (0.00, 0.12) | 2.59 | 0.0 |
| Qader, 2016 | Bangladesh | <19 years | eGFR <90 | eGFR <15 | 0.57 (0.45, 0.68) | 2.74 | |
| Yadav, 2016 | Nepal | <16 years | Not reported | Stages 4/5 | 1.00 (0.44, 1.00) | 1.58 | |
| Halle, 2017 | Cameroon | <18 years | eGFR <60 | eGFR <15 | 0.81 (0.57, 0.93) | 2.44 | |

*(Continued)*

**Table 4.** (*Continued*)

| Author | Country | Study age threshold | Study kidney function threshold | Late presentation definition used | Late presentation proportion, (95% CI) | % Weight* | Incidence (per million age related population) |
|---|---|---|---|---|---|---|---|
| Alparslan, 2017 | Turkey | <19 years | eGFR <60 | eGFR <15 | 0.36 (0.31, 0.43) | 2.82 | |
| Sandanala, 2018 | India | <19 years | eGFR <50 | eGFR <10 | 0.40 (0.17, 0.69) | 2.25 | |
| | | | | Random pooled effect size | 0.43 (0.34, 0.54) | 100.0 | Median incidence 2.1 (0.9, 3.9) |
| **Requirement for KRT within specified timeframe (denominator: incident ESKD/kidney failure population)** | | | | | | | |
| Eke, 1994 | Nigeria | <16 years | Not recorded | ESKD within 6 months of first visit | 1.00 (0.80, 1.00) | 6.43 | |
| Jander, 2006 | Poland | <19 years | Requiring KRT | KRT within 1 month of first visit | 0.21 (0.15, 0.27) | 13.84 | |
| USRDS report, 2008 | United States | <21 years | Requiring KRT | Not known to nephrologist | 0.35 (0.33, 0.38) | 15.25 | |
| van Stralen, 2010 | Europe-wide | <18 years | Requiring KRT | KRT within 1 month of first visit | 0.31 (0.26, 0.36) | 14.55 | |
| Boehm, 2010 | Austria | <18 years | Requiring KRT | KRT within 3 months of first visit | 0.24 (0.17, 0.33) | 12.97 | |
| Kennedy, 2012 | Australia | <18 years | Requiring KRT | KRT within 3 months of first visit | 0.30 (0.19, 0.44) | 10.62 | |
| Asinobi, 2014 | Nigeria | <18 years | Requiring KRT or death from ESKD | Not known to nephrologist | 0.64 (0.51, 0.76) | 11.01 | |
| Pruthi, 2016 | United Kingdom | <16 years | Requiring KRT | KRT within 3 months of first visit | 0.25 (0.23, 0.28) | 15.32 | |
| | | | | Random pooled effect size | 0.37 (0.29, 0.45) | 100.0 | |
| **eGFR-based threshold (denominator: incident ESKD/kidney failure population)** | | | | | | | |
| van Stralen, 2010 | Europe | <18 years | Requiring KRT | eGFR ≤20ml/min/1.73m² | 0.59 (0.54, 0.64) | 37.60 | |
| Kennedy, 2012 | Australia | | Requiring KRT | eGFR <30ml/min/1.73m² | 0.55 (0.40, 0.69) | 22.02 | |
| Pruthi, 2016 | United Kingdom | <16 years | Requiring KRT | eGFR <30ml/min/1.73m² | 0.71 (0.68, 0.73) | 40.38 | |
| | | | | Random pooled effect size | 0.63 (0.53, 0.73) | 100.0 | |
| **Time delay between symptom onset and diagnosis of CKD >3 months** | | | | | | | |
| Olowu, 2013 | Nigeria | <17 years | All stages | Symptoms >3 months before first presentation | 0.46 (0.38, 0.54) | 61.07 | |
| Odentunde, 2014 | Nigeria | <17 years | eGFR <60 | Symptoms >3 months before first presentation | 0.62 (0.52, 0.71) | 38.93 | |
| | | | | Random pooled effect size | 0.52 (0.46, 0.59) | 100.0 | |

Abbreviations: CI, Confidence Interval; CKD, Chronic Kidney Disease; CrCl Creatinine clearance; eGFR, estimated glomerular filtration rate (ml/min/1.73m²); ESKD, end-stage kidney disease (used synonymously with 'kidney failure'); KRT, renal replacement therapy; NAPRTCS, North American Pediatric Renal Trials and Collaborative Studies; USRDS, United States Renal Data System.

*Weight from random effects model

^Studies by Mong Hiep *et al.* (2008) and Huong *et al.* (2009) in combination report coverage of paediatric population for Vietnam; country-wide population data for year of study was therefore used to calculate incidence.

income and late presentation and was therefore not included in the final model. A lower aggregate eGFR at presentation strongly correlated with higher odds of late presentation on univariable analysis, however, due to limited studies with available data we were underpowered to incorporate in the final model. In the final model, country income and study setting were strongly associated with greater odds of late presentation. Exclusion of zero-event studies did not significantly alter effect estimates. Similarly, restricting analysis to most recent published data by centre or region did not change associations seen.

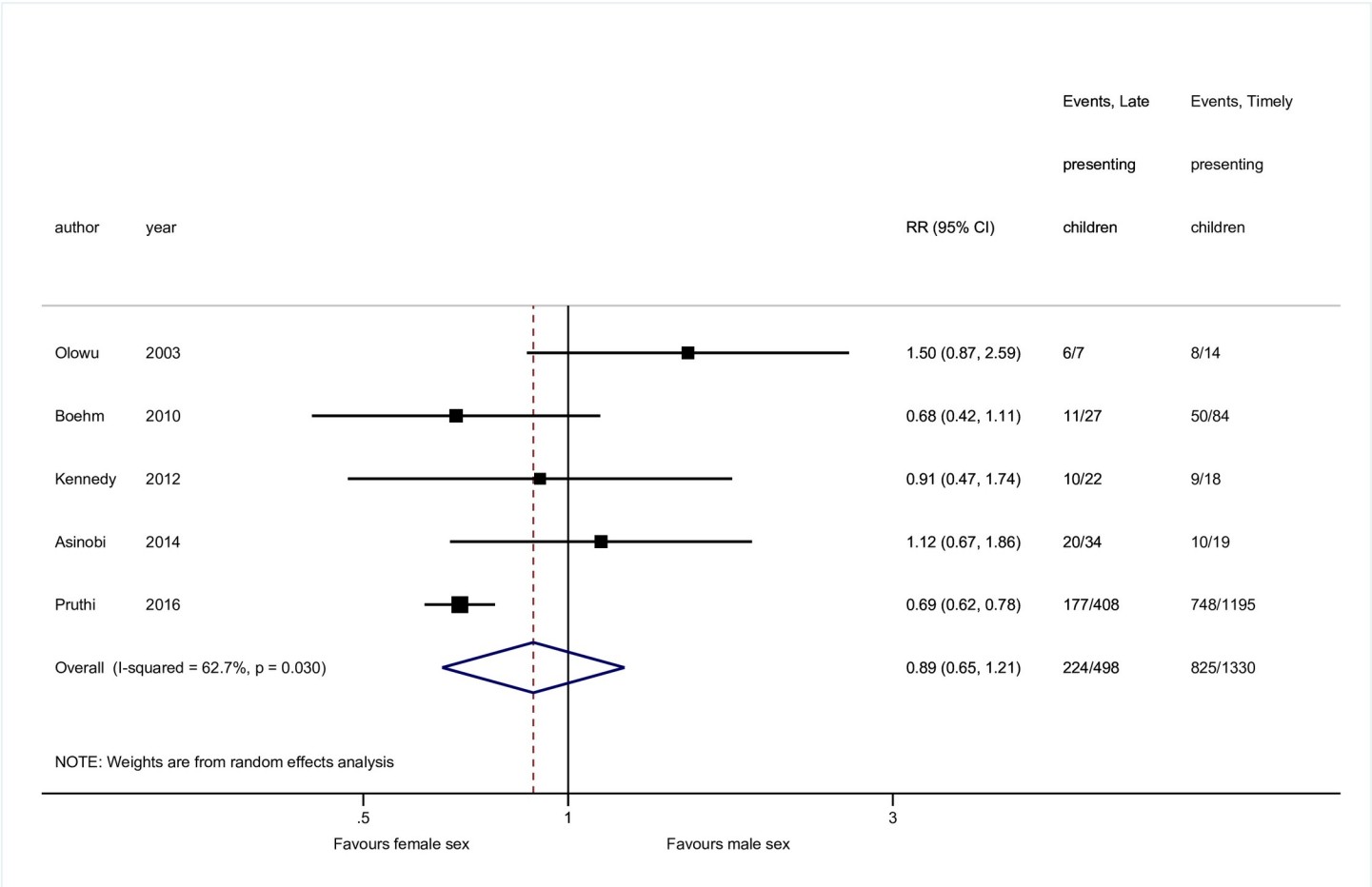

**Fig 2. Risk of late presentation by sex.** Abbreviations: CI, confidence interval; RR, Risk Ratio.

## Discussion

This systematic review extensively searched published and grey literature for epidemiological studies reporting the incidence and proportion of late presentation in children, irrespective of the main aim of the study. It demonstrated the term is commonly used to describe children with advanced disease, typically kidney failure, at initial presentation to nephrology services. Using this definition, the pooled proportion of children presenting late was 0.43 (95% CI 0.34–0.54); the median incidence was 2.1 (IQR 0.9–3.9) pmarp. Other definitions identified included early requirement for KRT and eGFR-based thresholds among the incident kidney failure cohort as well as time delays from symptom onset to presentation, and from secondary to specialist care. Meta-analysis highlighted that older age and non-congenital kidney disease were associated with greater risk of late presentation, however findings varied across studies. Considerable heterogeneity was noted between studies, which was partly explained by country income and study setting. Age of the study cohort was strongly associated with country income, suggesting national wealth may influence timing of access to nephrology care.

While other global systematic reviews have focused on access to KRT and outcomes for children with kidney failure [8,66], this is the first study to systematically quantify the burden of late presentation. Understanding this problem in detail is necessary for several reasons. First, we have identified late presentation to be a common global phenomenon for children

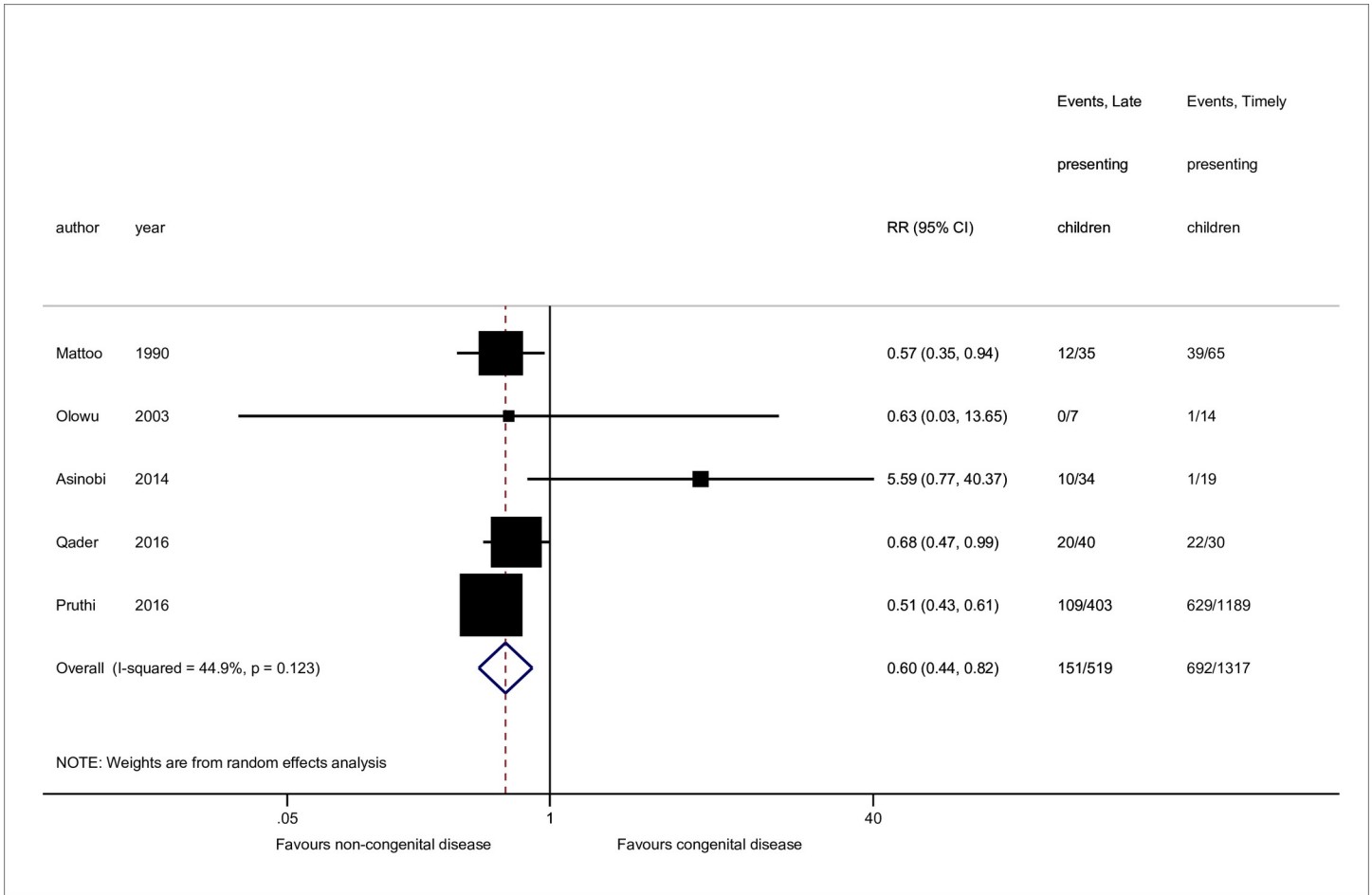

**Fig 3. Risk of late presentation by kidney disease type.** CAKUT denotes congenital anomalies of the kidneys and urinary tract. Studies either referenced 'CAKUT' disorders or sub-groups including obstructive uropathy, reflux nephropathy and/or renal hypo/dysplasia. Sub-groupings were combined for CAKUT total. Pruthi *et al.* reported missing disease data for 11 patients. Abbreviations: CI, confidence interval; RR, Risk Ratio.

diagnosed with CKD, with higher proportions noted among LMIC studies. This observation correlates with recent global estimates [67], suggesting that the burden of kidney disease is carried predominantly by resource-poor countries with limited access to KRT. This reinforces the urgent need for investment and intervention to support timely diagnosis and delay disease progression among these populations.

Second, identifying risk factors will enable the development of strategies to detect CKD earlier, as has been demonstrated by community-based adult programmes [68]. Although data were sparse, we identified greater risk of late presentation for children with non-congenital disease. Faster progression of CKD is seen among children with glomerular disorders compared with congenital disease which may explain presentation in advanced disease [9]. This may not be solely due to disease-type but mediated via proteinuria and hypertension [69]. Despite the male predominance often seen among congenital disorders, we did not identify an association between sex and late presentation. Any differences in CKD progression by sex may be mediated by disease-type rather than a direct effect of sex [70]. Older age was noted among late presenting children; both increasing age and the pubertal period have been linked to more rapid decline of kidney function which may in part explain this finding [29,71].

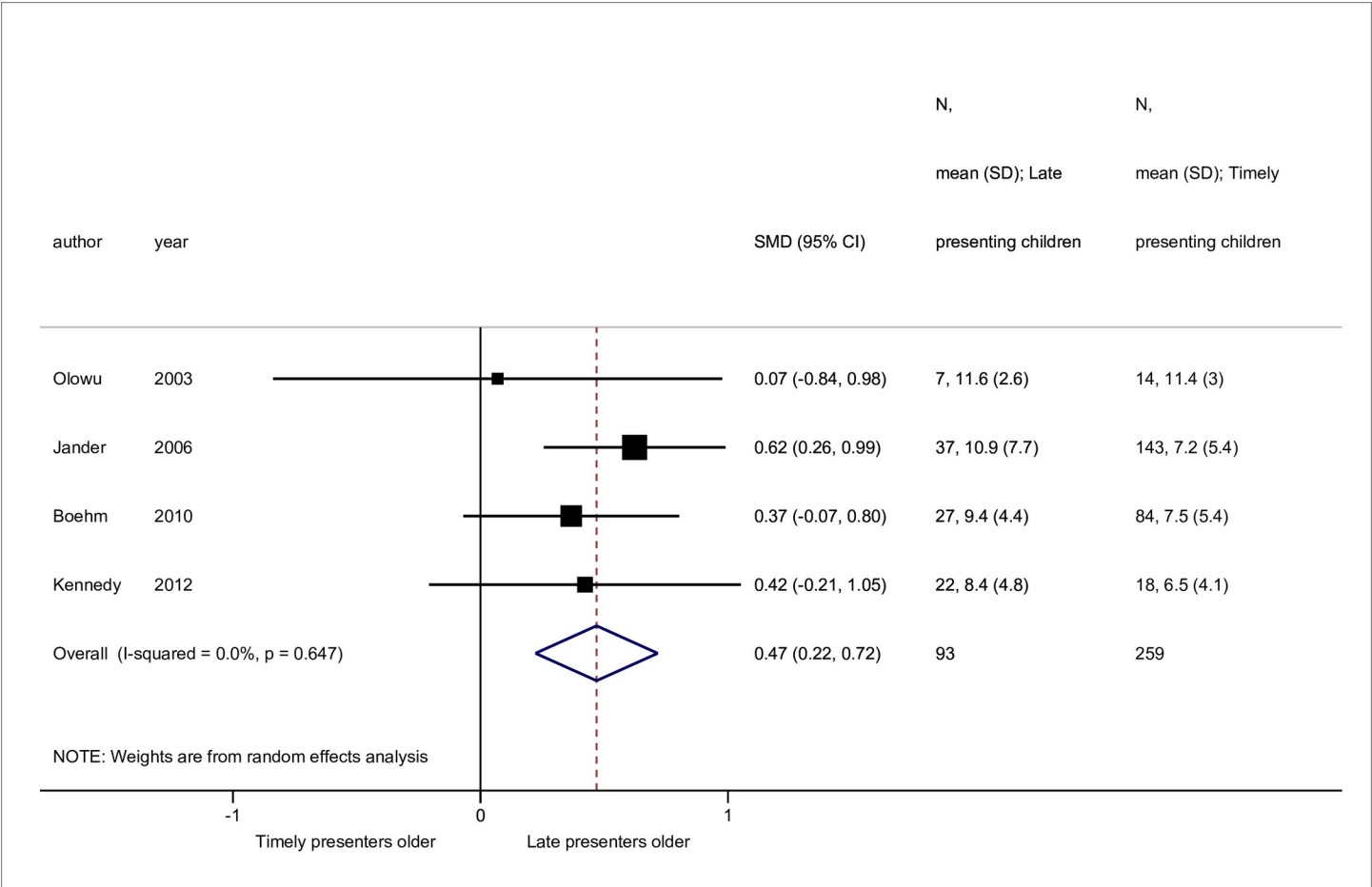

**Fig 4. Standardised mean differences in age at presentation between late versus timely presenters.** Abbreviations: SMD, standardised mean difference; CI, confidence interval.

The risk factors identified in this review are, overall, non-modifiable. What is not clear is whether there are modifiable factors that may serve as targets for intervention. Many authors alluded to factors that may hinder access to specialist healthcare, which included geographical remoteness from centre, socio-economic deprivation, and healthcare costs, however with the patient-level data available we were unable to substantiate these hypotheses. One Australian study identified an association between geographical remoteness and late presentation [58] however due to a lack of adjustment for potential confounders, our ability to interpret these results are limited.

Understanding the process of how late presenting children are recognised and reach specialist care, will help to identify opportunities for earlier detection. It will also clarify whether *late referral* to specialist care is implicated, as suggested by 2 studies included in this review [21,22]. Defining symptoms experienced pre-detection will also support strategies for intervention. Although many studies described symptoms at presentation, we were unable to extract these data by late presentation status. With as many as 50% of children reporting symptoms for several months prior to presentation [24,44], use of symptom type in combination with risk factors such as age could enable targeted use of CKD screening tests within healthcare settings.

**Table 5. Univariable and multivariable random effects meta-regression of late presentation of CKD (adjusted $R^2$ of multivariable model 52.6%).**

| Covariate | Univariable analysis | | | Multivariable analysis (*n* = 44 studies) | | |
|---|---|---|---|---|---|---|
| | Odds of late presentation | 95% Confidence interval | *P* value | Odds of late presentation | 95% Confidence interval | *P* value |
| **Group two definition** *(c.f. group one)* | 0.77 | 0.32, 1.87 | 0.56 | - | - | - |
| **Study from high-income country** *(c.f. low- or middle-income income country)* | 0.33 | 0.18, 0.60 | 0.001 | 0.41 | 0.24, 0.71 | 0.002 |
| **Multi-centre design** *(c.f. single-centre)* | 0.68 | 0.34, 1.38 | 0.28 | - | - | - |
| **Hospital admissions** *(c.f. all paediatric nephrology services)* | 3.93 | 1.91, 8.05 | <0.001 | 3.01 | 1.56, 5.80 | 0.002 |
| **GFR-based inclusion threshold** *(c.f. creatinine-based threshold)* | 0.57 | 0.14, 2.40 | 0.43 | - | - | - |
| **Study conducted from 2001 onwards** *(c.f. pre-2001, median start year)* | 1.35 | 0.68, 2.68 | 0.38 | - | - | - |
| **Large sample size (≥140 participants)** *(c.f. <140 participants, median sample size)* | 0.64 | 0.33, 1.24 | 0.18 | - | - | - |
| **Study summary age** *Median/mean age of study cohort (per year increase)* | 1.23 | 1.11, 1.36 | <0.001 | - | - | - |
| **GFR at presentation** *Median/mean estimated glomerular filtration rate of study cohort (per ml/min/1·73m² increase)* | 0.96 | 0.94, 0.99 | 0.01 | - | - | - |

Abbreviations: CKD, Chronic Kidney Disease; GFR, glomerular filtration rate. *Due to limited number of studies (*n* = 15), we were underpowered to include GFR at presentation in the final multivariable model.

Third, appraising the literature provides an opportunity to highlight variations in study methods and to standardise approaches for future research. Since the publication of the National Kidney Foundation's Kidney Disease Outcomes Quality Initiative guideline in 2002 [72] and its adoption by KDIGO in 2005 [73], there has been an internationally-recognised classification system for children >2 years which uses a threshold of <60ml/min/1.73m² to define CKD. Many group 1 studies referenced these guidelines in either defining or staging CKD (54.1%), however only 10 studies(27.0%) used an eGFR threshold of <60ml/min/1.73m² for study inclusion, with others using narrower or broader CKD definitions. Variation in functional thresholds to define CKD contributed to the heterogeneity seen, as evidenced by meta-regression findings. Standardising functional thresholds using international guidance as well as achieving consensus on what constitutes a 'paediatric' population will enable future epidemiological research to be directly comparable across nations.

This study has several strengths. We extensively searched the literature for studies describing or referencing late presentation of kidney disease, regardless of the study aim. We employed a comprehensive search strategy that included forward- and backward-citation searching and sought to be inclusive of all studies, including those with small sample sizes or different definitions of late presentation. There are however limitations. To ensure incident populations were captured, studies describing prevalent populations were excluded; it is possible that these studies may have contained information relevant to our analysis. Second, a systematic review is only as good as the quality and content of the included studies. Those included in this review were generally small, single-centre, descriptive studies derived from heterogenous clinical populations and are therefore at risk of bias. Given the specialist-based populations used to determine study cohorts, estimates generated are likely to represent the 'tip of the iceberg', with severe CKD being over-represented. Furthermore, our analyses sought to combine studies with differing inclusion criteria and age thresholds, although efforts to formally quantify heterogeneity using meta-regression were employed to counteract this problem. It must also be recognised that country income and age of the study cohort represent aggregate data and may not reflect true associations on an individual level.

In conclusion, late presentation, most commonly defined as first presentation to paediatric nephrology care with kidney failure, is a global phenomenon affecting a large proportion of children attending specialist services and is more common in older children and those with non-congenital kidney disease. Studies from LMIC report higher proportions of late presentation than HIC studies, highlighting an urgent need for interventions to reduce the burden of late detection in settings with limited access to KRT. International consensus on a definition for late presentation, as well as age ranges to be included within paediatric studies, will standardise reporting of epidemiological studies and facilitate comparison of risk associations across populations. Research is required to explore non-congenital disease associations and consider other factors implicated in late presentation, such as socio-economic deprivation. Due to the heterogeneity in estimates observed, a tailored approach is required to determine key individual, healthcare, and region-specific factors contributing to the problem.

## Supporting information

**S1 Checklist.**
(DOC)

**S1 Fig. MEDLINE search strategy.**
(TIF)

**S2 Fig. List of grey literature databases.**
(TIF)

**S3 Fig. Funnel plot assessment of publication bias.**
(TIF)

**S1 Table. Modified Newcastle-Ottawa score.**
(DOCX)

**S2 Table. Description of studies included in systematic review.** Abbreviations: AKI, Acute Kidney Injury; ATN, acute tubular necrosis; CrCl, creatinine clearance; CRF, chronic renal failure; CKD, chronic kidney disease [Note: Terminology used reflects that of study. CRF and CKD terms used interchangeably]; ESRF, End-Stage Renal Failure; ESRD, End-Stage Renal Disease [Note: Terminology used reflects that of study. ESRF and ESRD terms used interchangeably and are synonymous with 'kidney failure']; eGFR, estimated Glomerular Filtration Rate; NAPRTCS, North American Pediatric Trials and Collaborative Studies; NR, not recorded; PN, Paediatric Nephrology; RRT, Renal Replacement Therapy [Note terminology used reflects that of study and is used interchangeably with Kidney Replacement Therapy]; SD, standard deviation; SES, socio-economic status; USRDS, United States Renal Data System. *Numbers of excluded patients are provided where available.
(DOCX)

## Acknowledgments

We acknowledge Alison Richards, Information specialist, for her support in developing the search strategy. We thank Dr Jelena Savović and Dr Vincent Cheng for their respective help with Serbian and Chinese translations and Profs Jonathan Sterne and Julian Higgins for their statistical support. We would also like to thank the University of Bristol inter-library loan service.

## Author Contributions

**Conceptualization:** Lucy Plumb, Fergus J. Caskey, Manish D. Sinha, Yoav Ben-Shlomo.

**Data curation:** Lucy Plumb.

**Formal analysis:** Lucy Plumb, Fergus J. Caskey, Manish D. Sinha, Yoav Ben-Shlomo.

**Funding acquisition:** Lucy Plumb, Fergus J. Caskey, Manish D. Sinha, Yoav Ben-Shlomo.

**Investigation:** Lucy Plumb, Emily J. Boother, Yoav Ben-Shlomo.

**Methodology:** Lucy Plumb, Fergus J. Caskey, Manish D. Sinha, Yoav Ben-Shlomo.

**Supervision:** Fergus J. Caskey, Manish D. Sinha, Yoav Ben-Shlomo.

**Writing – original draft:** Lucy Plumb, Emily J. Boother, Fergus J. Caskey, Manish D. Sinha, Yoav Ben-Shlomo.

**Writing – review & editing:** Lucy Plumb, Emily J. Boother, Fergus J. Caskey, Manish D. Sinha, Yoav Ben-Shlomo.

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
