## [Decision Letter · Decision Letter 0]

10 Nov 2020

PONE-D-20-29742

The incidence of and risk factors for late presentation of childhood chronic kidney disease: a systematic review and meta-analysis

PLOS ONE

Dear Dr. Plumb,

Thank you for submitting your manuscript to PLOS ONE. After careful consideration, we feel that it has merit but does not fully meet PLOS ONE’s publication criteria as it currently stands. Therefore, we invite you to submit a revised version of the manuscript that addresses the points raised during the review process.

We look forward to receiving your revised manuscript.

Kind regards,

Jennifer A Hirst, DPhil

Academic Editor

PLOS ONE

Journal Requirements:

2. We note that the original search was performed in 10 September 2019. Please discuss whether relevant literature has been published in the interim that would be expected to affect the results of the meta-analysis.

3. At this time, we ask that you please provide the full search strategy and search terms for at least one database used as Supplementary Information.

"L.P reports grants from National Institute for Health Research and grants from Kidney Research UK during the conduct of the study.

F.J.C reports grants from NIHR, grants from Kidney Research UK, and personal fees from Baxter outside the submitted work.

M.D.S acknowledges financial support from the Department of Health via the National Institute for Health Research (NIHR) comprehensive Biomedical Research Centre and Clinical Research Facilities awards to Guy’s and St Thomas’ NHS Foundation Trust in partnership with King’s College London and King’s College Hospital NHS Foundation Trust.

Y.B-S is partly funded by National Institute for Health Research Applied Research Collaboration West (NIHR ARC West) at University Hospitals Bristol NHS Foundation Trust. "

Reviewers' comments:

Reviewer's Responses to Questions

**Comments to the Author**

1. Is the manuscript technically sound, and do the data support the conclusions?

Reviewer #1: Yes

Reviewer #2: Yes

2. Has the statistical analysis been performed appropriately and rigorously? 

Reviewer #1: Yes

Reviewer #2: Yes

3. Have the authors made all data underlying the findings in their manuscript fully available?

Reviewer #1: Yes

Reviewer #2: Yes

4. Is the manuscript presented in an intelligible fashion and written in standard English?

Reviewer #1: Yes

Reviewer #2: Yes

5. Review Comments to the Author

Reviewer #1: I thoroughly enjoyed reviewing this paper and commend the author group on their hard work. Some comments:

The conclusion that there is a higher proportion of late presentation in studies of hospitals compared to studies of specialist services is expected given a more dilute population of patients would present to paediatric nephrology services with stable CKD whereas patients with critical illness are far more likely to be admitted to hospital.

Nonetheless, discussion of the factors around timely recognition of CKD and appropriate access to healthcare as factors that contribute to late presentation is important. It would be important to see how socio-economic factors surrounding caregivers of children with CKD impact on late presentation - I accept that extracting these individual factors would be difficult, however, I also believe that it would make your conclusions more powerful (as per statement on Page 21 “country income and age of the study cohort represent aggregate data and may not reflect true associations on an individual level.". This is touched upon in some detail, linked to other factors like geographic remoteness and LMIC/HIC countries. Studies like Francis et al (https://link.springer.com/article/10.1007/s00467-015-3279-z) would strengthen this discussion to explore the individual factors further - after all, having caregivers in the middle SES quintile in a LIC vs caregivers in the middle SES quintile of a HIC would impact differently on the child.

The other concern of note is the delineation of the included studies into five main categories of ‘late presentation’. In some instances, groups 1-3 are very similar. The authors even refer to “Another definition noted in a prospective Nigerian single-centre study describes[ing] (sic) LP as the development of kidney failure within 6 months of first presentation” - this definition sounds like it would be included in Group 2?

It was unclear to me why only publications from 1990 would be included? While this would increase the amount of single-centre studies, it is important to be inclusive and complete in the original search. One could run a sensitivity analysis on studies pre- and post-1990 to delineate potential differences in bias. Otherwise, a thorough search strategy.

Overall, a well-executed, technically sound systematic review and meta-analysis. I would recommend minor revisions.

Reviewer #2: The authors of this manuscript conducted a systematic review and meta-analysis to determine the incidence of late presentation of chronic kidney disease (CKD) in children and its associated risk factors. After an in-depth and a well-researched review with good statistical analysis, they found a median incidence of 2.1 per million age-related population (pmrap), with non-congenital kidney diseases and older age as related risk factors. Undoubtedly, late stage of CKD, especially end-stage renal failure (ESRF) or end-stage kidney disease (ESKD), contributes to the global health burden in children. This underscores the need for a comprehensive data which reports the extent of this non-communicable disease. However, there are few specific concerns that should be addressed by the authors to improve the manuscript.

Specific comments

1. Abstract- Under Background, I think the term ‘kidney failure’ in line 25 should read ‘end-stage renal failure’ or ‘end-stage kidney disease’ since CKD progresses from stages 1 to 4 CKD and ends up in stage 5 CKD or end-stage renal failure. This observation also applies for ‘kidney failure’ mentioned in line 35 (under Results), although you stated in Table 2 that both terms are synonymous..

2. Introduction- Similarly, I suggest you replace the term ‘kidney failure’ in line 57 (and elsewhere in the manuscript) with ‘end-stage renal failure’ or ‘end-stage kidney disease’ and ‘kidney replacement therapy’ (KRT) with ‘renal replacement therapy’ (RRT). In lines 69 and 74 (and elsewhere in the manuscript), using the abbreviation ‘LP’ for late presentation is unnecessary and could be deleted more so when the same abbreviation also refers to one of the authors! (Line 115).

3. Materials and Methods- Table 1 on the systematic review criteria appears redundant and could be deleted. Presenting the information in prose form may suffice.

4. Discussion- In line 354, do you mean ‘guidance’ or ‘guideline’? Clarify and correct.

6. PLOS authors have the option to publish the peer review history of their article (what does this mean?). If published, this will include your full peer review and any attached files.

Reviewer #1: No

Reviewer #2: **Yes: **Samuel Uwaezuoke

---

## [Author Response · Author response to Decision Letter 0]

9 Dec 2020

We would like to thank the editors for consideration of our manuscript and the reviewers for their insightful and detailed comments.

In response to editorial comments:

- We have checked that our manuscript aligns with PLOS ONE’s style requirements. Please do let us know if there are any further modifications to make. 

- We have repeated our searches, restricted to published data available in 2020 and have not found any studies that may alter our findings. 

- We highlight that an example MEDLINE search strategy is included as supplemental figure 1, in accordance with the PRISMA checklist. 

- We have added a comment about competing interests to our revision cover letter.

Below are responses to each reviewer point raised, which we hope are satisfactory. The main body of the manuscript is now 3580 words long. 

Reviewer #1: I thoroughly enjoyed reviewing this paper and commend the author group on their hard work. Some comments:

The conclusion that there is a higher proportion of late presentation in studies of hospitals compared to studies of specialist services is expected given a more dilute population of patients would present to paediatric nephrology services with stable CKD whereas patients with critical illness are far more likely to be admitted to hospital.

Nonetheless, discussion of the factors around timely recognition of CKD and appropriate access to healthcare as factors that contribute to late presentation is important. It would be important to see how socio-economic factors surrounding caregivers of children with CKD impact on late presentation - I accept that extracting these individual factors would be difficult, however, I also believe that it would make your conclusions more powerful (as per statement on Page 21 “country income and age of the study cohort represent aggregate data and may not reflect true associations on an individual level.". This is touched upon in some detail, linked to other factors like geographic remoteness and LMIC/HIC countries. Studies like Francis et al (https://link.springer.com/article/10.1007/s00467-015-3279-z) would strengthen this discussion to explore the individual factors further - after all, having caregivers in the middle SES quintile in a LIC vs caregivers in the middle SES quintile of a HIC would impact differently on the child.

We thank the reviewer for this comment and agree that socioeconomic factors of the family may be implicated in late presentation of childhood chronic kidney disease. We attempted to extract these data however in many cases these data were not available stratified by late presentation status or stage at presentation. We identified one study that quantified low socioeconomic status although the measure used was not included within the study methodology:

[Lines 268-271] “Two studies provided unadjusted data for other risk factors including low socio-economic status (SES) and geographical remoteness respectively. Michael and Gabreil[32] described a cohort of children all presenting with an eGFR of <25ml/min/1.73m2 of which 87.5% were of low SES, although the definition and measure of SES used was unavailable.”

Within our data extraction proforma, we included a free text field whereby any comments regarding factors associated with late presentation/referral that were not substantiated were included. We have now included a comment about this qualitative analysis within the manuscript:

[Lines 273-277] “Qualitative text analysis was undertaken to identify features described but not substantiated that may be linked to late presentation; these included low socioeconomic status and costs associated with healthcare[21,35,38,46] living in remote areas[35], delayed referral to tertiary care [47], ‘traditional habits’ of first seeking healthcare from traditional healers and a lack of specialist education in local hospitals[39]”.

The other concern of note is the delineation of the included studies into five main categories of ‘late presentation’. In some instances, groups 1-3 are very similar. The authors even refer to “Another definition noted in a prospective Nigerian single-centre study describes[ing] (sic) LP as the development of kidney failure within 6 months of first presentation” - this definition sounds like it would be included in Group 2?

We thank the reviewer for this observation. The study contained within definition 6, does describe all children presenting to nephrology care who develop kidney failure within a short time-frame, and therefore is comparable to studies within group 2. We have rectified this and re-run analyses that include group 2 studies. The descriptive text for group two has also been updated. We apologise for the confusion. 

It was unclear to me why only publications from 1990 would be included? While this would increase the amount of single-centre studies, it is important to be inclusive and complete in the original search. One could run a sensitivity analysis on studies pre- and post-1990 to delineate potential differences in bias. Otherwise, a thorough search strategy.

We thank the reviewer for their suggestion. We chose not to commence our search prior to 1990 for two reasons. Firstly, we anticipated a large number of studies describing cohorts of children attending specialist nephrology care and therefore sought to limit the search time-frame so as to conduct a thorough yet manageable search. Secondly, many paediatric nephrology units and respective training programmes were in their infancy or being established prior to 1990, as described in this article (https://www.nature.com/articles/pr2002256.pdf?origin=ppub) – we therefore aimed to focus our search on studies from a relatively recent time-period, to reflect modern specialist healthcare access globally. We have added now added a rationale for our search time-frame to our methods:

[Lines 101-104] “Studies published between 1 January 1990 and 10 September 2019 which were available in paper or electronic form were included: this time-frame was chosen to adequately capture studies from established paediatric nephrology centres and reflect modern specialist healthcare access globally.“

We also performed a sensitivity analysis, using only the most recent data from a centre or region where older study data had also been published. This did not significantly alter the overall proportion of children presenting late or associations noted within the meta-regression. We have now incorporated these findings into the manuscript.

[Lines 245-246] “Additionally, restricting data to the most recent study from each centre or region did not significantly alter findings.”

[Lines 299-300] “Similarly, restricting analysis to most recent published data by centre or region did not change associations seen.”

Overall, a well-executed, technically sound systematic review and meta-analysis. I would recommend minor revisions.

Reviewer #2: The authors of this manuscript conducted a systematic review and meta-analysis to determine the incidence of late presentation of chronic kidney disease (CKD) in children and its associated risk factors. After an in-depth and a well-researched review with good statistical analysis, they found a median incidence of 2.1 per million age-related population (pmrap), with non-congenital kidney diseases and older age as related risk factors. Undoubtedly, late stage of CKD, especially end-stage renal failure (ESRF) or end-stage kidney disease (ESKD), contributes to the global health burden in children. This underscores the need for a comprehensive data which reports the extent of this non-communicable disease. However, there are few specific concerns that should be addressed by the authors to improve the manuscript.

Specific comments

1. Abstract- Under Background, I think the term ‘kidney failure’ in line 25 should read ‘end-stage renal failure’ or ‘end-stage kidney disease’ since CKD progresses from stages 1 to 4 CKD and ends up in stage 5 CKD or end-stage renal failure. This observation also applies for ‘kidney failure’ mentioned in line 35 (under Results), although you stated in Table 2 that both terms are synonymous.

This manuscript has been written using recommendations from the Kidney Diseases Improving Global Outcomes (KDIGO) 2020 consensus conference (https://www.nature.com/articles/s41581-020-0290-9), which advocates use of the term ‘kidney failure’ in place of ‘end-stage kidney disease’. For this reason, kidney failure and kidney replacement therapy are used throughout the manuscript. We have now referenced this within the methodology and hope this will be acceptable. We have also checked the manuscript to ensure all kidney-related terms align where possible to these recommendations.

[Lines 124-126] “The terminology contained within this report aligns with recommendations from Kidney Disease Improving Global Outcomes (KDIGO) consensus conference on nomenclature for kidney function and disease[16], although reference to study-specific terminology is made.”

2. Introduction- Similarly, I suggest you replace the term ‘kidney failure’ in line 57 (and elsewhere in the manuscript) with ‘end-stage renal failure’ or ‘end-stage kidney disease’ and ‘kidney replacement therapy’ (KRT) with ‘renal replacement therapy’ (RRT). In lines 69 and 74 (and elsewhere in the manuscript), using the abbreviation ‘LP’ for late presentation is unnecessary and could be deleted more so when the same abbreviation also refers to one of the authors! (Line 115).

Please see the comment above regarding nomenclature used – we hope our rationale for this is acceptable to the reviewer and editorial team. We thank you also for the comment regarding the abbreviation of late presentation. We have now removed this from the manuscript. 

3. Materials and Methods- Table 1 on the systematic review criteria appears redundant and could be deleted. Presenting the information in prose form may suffice. 

We have now removed this table from the manuscript and added information to the text. 

4. Discussion- In line 354, do you mean ‘guidance’ or ‘guideline’? Clarify and correct. 

Our apologies for this error- this has now been corrected to ‘guideline’.

---

## [Editor Report · Decision Letter 1]

16 Dec 2020

The incidence of and risk factors for late presentation of childhood chronic kidney disease: a systematic review and meta-analysis

PONE-D-20-29742R1

Dear Dr. Plumb,

We’re pleased to inform you that your manuscript has been judged scientifically suitable for publication and will be formally accepted for publication once it meets all outstanding technical requirements.

Kind regards,

Jennifer A Hirst, DPhil

Academic Editor

PLOS ONE
---

## [Editor Report · Acceptance letter]

21 Dec 2020

PONE-D-20-29742R1 

The incidence of and risk factors for late presentation of childhood chronic kidney disease: a systematic review and meta-analysis 

Dear Dr. Plumb:

I'm pleased to inform you that your manuscript has been deemed suitable for publication in PLOS ONE. Congratulations! Your manuscript is now with our production department. 

Kind regards, 

on behalf of

Dr. Jennifer A Hirst 

Academic Editor

PLOS ONE